# Sorting of a multi-subunit ubiquitin ligase complex in the endolysosome system

**Xi Yang, Felichi Mae Arines, Weichao Zhang, Ming Li\***

Department of Molecular, Cellular and Developmental Biology, University of Michigan, Ann Arbor, United States

**Abstract** The yeast Dsc E3 ligase complex has long been recognized as a Golgi-specific protein ubiquitination system. It shares a striking sequence similarity to the Hrd1 complex that plays critical roles in the ER-associated degradation pathway. Using biochemical purification and mass spectrometry, we identified two novel Dsc subunits, which we named as Gld1 and Vld1. Surprisingly, Gld1 and Vld1 do not coexist in the same complex. Instead, they compete with each other to form two functionally independent Dsc subcomplexes. The Vld1 subcomplex takes the AP3 pathway to reach the vacuole membrane, whereas the Gld1 subcomplex travels through the VPS pathway and is cycled between Golgi and endosomes by the retromer. Thus, instead of being Golgi-specific, the Dsc complex can regulate protein levels at three distinct organelles, namely Golgi, endosome, and vacuole. Our study provides a novel model of achieving multi-tasking for transmembrane ubiquitin ligases with interchangeable trafficking adaptors.
DOI: https://doi.org/10.7554/eLife.33116.001

## Introduction

Ubiquitin-dependent protein down-regulation and quality control are important for maintaining the integrity of all organelles (*Arvan et al., 2002*; *Wang et al., 2011*). Besides the well-characterized ER protein quality control system (*Ellgaard and Helenius, 2003*; *Ruggiano et al., 2014*), other organelles such as the plasma membrane (*Babst, 2014*; *Okiyoneda et al., 2010*; *Zhao et al., 2013*), Golgi (*Dobzinski et al., 2015*; *Reggiori and Pelham, 2002*), endosomes (*Fukuda et al., 2006*; *Léon et al., 2008*; *Nakamura et al., 2005*), peroxisome (*Platta et al., 2009*; *Williams et al., 2008*), and mitochondria (*Narendra et al., 2008*), either contain integral membrane E3 ligases, or have adaptor proteins to recruit cytosolic E3 ligases for the removal of damaged organelle proteins or the down-regulation of unnecessary proteins (*Hettema et al., 2004*; *Lin et al., 2008*). Focusing on the down-regulation of lysosomal membrane proteins, we recently identified two independent E3 ligase systems on the budding yeast vacuole (yeast lysosome) membrane, each governing a different subset of vacuolar membrane proteins. The Ssh4-Rsp5 system selectively ubiquitinates a vacuolar lysine transporter Ypq1 to down-regulate the lysine import activity after lysine withdrawal (*Li et al., 2015b*; *Sekito et al., 2014*), whereas $Zn^{2+}$ withdrawal results in the selective ubiquitination of a $Zn^{2+}$ importer Cot1 by the Dsc E3 ligase complex (*Li et al., 2015a*). Interestingly, Rsp5 can work together with the Dsc complex to down-regulate a $Zn^{2+}$ exporter Zrt3 when excessive $Zn^{2+}$ is present in the cytoplasm (*Li et al., 2015a*). These ubiquitinated membrane proteins will be directly internalized into the vacuole lumen by the ESCRT machinery for degradation (*Zhu et al., 2017*).

Originally discovered in fission yeast, the *S. pombe* Dsc complex contains six components, including Tul1, Dsc2, Dsc3, Dsc4, Ubx3, and the AAA+ ATPase Cdc48 (*Stewart et al., 2012*, *2011*). These components, with the exception of Dsc4, also exist in budding yeast (*Dobzinski et al., 2015*; *Li et al., 2015a*; *Tong et al., 2014*). Strikingly, most Dsc components share sequence similarity to the Hrd1 E3 ligase complex, a key player in ER protein quality control. Tul1 is a multi-spanning membrane RING domain E3 ligase that is related to Hrd1. Other components, including Dsc2, Dsc3,

**\*For correspondence:**
mlium@umich.edu

**Competing interests:** The authors declare that no competing interests exist.

**eLife digest** Proteins perform many tasks and, to remain healthy, each cell must ensure that its proteins are in good condition and present at the right levels. Plants, animals and fungi all largely deal with damaged, or otherwise unneeded, proteins by tagging them with a small marker called ubiquitin. The tagged proteins are then rapidly destroyed, which prevents them from harming the cells.

Enzymes known as E3 ligases attach ubiquitin to proteins. Yet, the number of E3 ligases is dwarfed by the number of proteins modified with ubiquitin. For instance, humans have approximately 20,000 different proteins, about one third of which are found in or on cell membranes. However, there are only around 600 E3 ligases, and only about 50 of them are associated with cell membranes. This is further complicated by the fact that proteins are also present in distinct compartments within the cell.

The Dsc complex, for example, is an E3 ligase from yeast that is found within a compartment of the cell known as the Golgi. It was thus expected to only attach ubiquitin to Golgi proteins. Yet some recent studies showed that the Dsc complex could also tag proteins present in two other compartments of yeast cells: the endosome and vacuole. How can the Dsc complex act on proteins in three distinct compartments?

The Dsc complex is actually made from multiple proteins, and Yang et al. now report two new protein components. Biochemical and genetic tools showed that these two proteins do not co-exist in the same Dsc complex. Instead, they compete with each other to form two different kinds of Dsc complexes, which Yang et al. refer to as subcomplexes.

Further work showed that the two new proteins determine the route taken by the Dsc complex along the cell's protein transport pathway. One subcomplex is transported to the vacuole and the other cycles between the Golgi and endosomes. Thus, by changing just one component, the Dsc complex can be sent to different locations within the cell.

These findings describe a new mechanism that enables E3 ligases to multi-task on a wide range of proteins, even across distinct compartments of the cell. Future work will determine whether plant and animal cells also use a similar strategy. Since defects in protein quality control contribute to many human diseases, such as Alzheimer's and Parkinson's disease, working out how E3 ligases work is important for the field of biomedicine.

DOI: https://doi.org/10.7554/eLife.33116.002

Ubx3, are homologous to Der1, Usa1, and Ubx2 of the Hrd1 complex, respectively (*Stewart et al., 2012*, *2011*). Furthermore, both complexes contain the same AAA+ ATPase Cdc48. The striking similarity suggests the Dsc complex might play a role in protein quality control at the downstream organelles of the secretory pathway.

Probably the biggest controversy about the Dsc complex is its subcellular localization. In *S. pombe*, the Dsc complex has been shown to be critical to the proteolytic activation of the sterol regulatory element binding protein (SREBP) transcription factor, which is a Golgi membrane protein. Consistently, fission yeast Dsc complex has been shown to localize to the Golgi (*Burr et al., 2017*; *Stewart et al., 2011*). In *S. cerevisiae*, it is also generally accepted that the Dsc complex is a Golgi-specific E3 ligase complex. Tul1 was initially identified as a Golgi protein quality control E3 ligase through its ability to recognize and ubiquitinate an artificial folding mutant of Pep12, which is cycled between Golgi and endosomes (*Reggiori and Pelham, 2002*). Recently, it has been shown that the Dsc complex is also responsible for the ubiquitination and degradation of another Golgi membrane protein Yif1, after either amino acid starvation or rapamycin treatment (*Dobzinski et al., 2015*). However, the *Arabidopsis* homologue of Tul1, FLY1, is predominantly localized to the late endosome, instead of the Golgi (*Voiniciuc et al., 2013*). Furthermore, Graham and colleagues recently demonstrated that the budding yeast Tul1 participates in the ubiquitination and recycling of an exocytic v-SNARE Snc1 at the early endosome (*Xu et al., 2017*). Lastly, we observed that Dsc complex is responsible for the down-regulation of some vacuole membrane proteins in budding yeast (*Li et al., 2015a*). How can a Golgi E3 ligase complex ubiquitinate a vacuole membrane protein such as Cot1?

In this study, we resolved this controversy by identifying two new components of the Dsc complex, which we named as Gld1 (Golgi/endosome Localized Dsc protein 1) and Vld1 (Vacuole Localized Dsc protein 1). Gld1 and Vld1 are two similar tetra-spanning membrane proteins that compete with each other to form functionally independent Dsc subcomplexes at the ER. The Gld1-containing subcomplex takes the VPS (vacuolar protein sorting) pathway for its localization and is cycled between endosomes and Golgi by the retromer complex. In contrast, the Vld1-containing subcomplex travels through the AP3 pathway to the vacuole membrane. Together, this novel mechanism enables the cell to achieve protein regulation and probably quality control at three distinct organelles, namely Golgi, endosomes, and the vacuole, using just one RING domain E3 ligase Tul1. We propose that plant and mammalian cells might use a similar strategy to target their membrane-residing E3 ligases to different organelles in order to expand their substrate repertoire.

## Results

### Targeting pathways utilized by the Dsc complex

Two independent pathways have been identified to deliver proteins from the Golgi to vacuole (*Figure 1A*). The VPS pathway transports vacuolar proteases such as CPY, Pep4 and Prb1 to the vacuole lumen via the intermediate endosomal compartments marked by the Pep12 SNARE protein (*Bowers and Stevens, 2005*). It is also used to deliver ubiquitinated membrane cargoes to the vacuole lumen for degradation. Furthermore, some vacuolar membrane proteins, such as Vph1 and Ssh4, also utilize the VPS pathway to reach the vacuole surface (*Zhu et al., 2017*). The VPS pathway can be blocked by deletion of *PEP12*, or the cargo can be trapped in an aberrant endosome (i.e. the class E compartment) by the deletion of genes encoding ESCRT machinery (*Bowers and Stevens, 2005*; *Odorizzi et al., 1998a*; *Zhu et al., 2017*).

As an independent targeting mechanism, the AP3 pathway transports a subset of the vacuolar membrane proteins such as alkaline phosphatase (ALP) from Golgi to the vacuole (*Figure 1A*) (*Cowles et al., 1997*). These membrane proteins typically contain an acidic di-leucine targeting motif (D/EXXXLL, where X can be any amino acid), which can be recognized at the late Golgi by the AP3 adaptor complex for sorting into carrier vesicles that then directly target and fuse with the vacuole (*Cowles et al., 1997*). Deletion of the AP3 complex leads to an accumulation of AP3 cargoes at the Golgi and forces them to traffic to the vacuole membrane via the VPS pathway (*Cowles et al., 1997*; *Li et al., 2015b*; *Llinares et al., 2015*; *Odorizzi et al., 1998b*).

Previously, we reported that the Dsc complex has three distinct subcellular localizations, including Golgi, endosomes, and vacuole. At the steady state, about 60% of the Dsc complex localizes to the vacuole membrane, whereas the rest appears to be on punctae that co-localize with the Golgi and endosomes (*Li et al., 2015a*). This multi-localization pattern is very interesting among membrane proteins in the endomembrane trafficking pathway. At steady state, most endomembrane proteins are either localized to the vacuole membrane, or to punctae that include Golgi and endosomes. For example, Vph1, a $V_O$ subunit of the vATPase complex localizes to the vacuole membrane (*Manolson et al., 1992*), while its isoform, Stv1, localizes to Golgi and endosomes (*Manolson et al., 1994*). As another example, Ssh4 and Ear1, two homologous adaptor proteins for the E3 ligase Rsp5, localize to the vacuole and Golgi/endosomes, respectively (*Léon et al., 2008*; *Li et al., 2015b*). Furthermore, to maintain their punctae localization, some Golgi and endosomal membrane proteins are constantly recycled by the retromer complex (*Burd and Cullen, 2014*; *Hettema et al., 2003*; *Seaman et al., 1997*). Only in retromer mutants do these Golgi/endosome proteins mislocalize to the vacuole membrane (*Burd and Cullen, 2014*; *Strochlic et al., 2007*). Then, how does the Dsc complex achieve three distinct subcellular localizations in wild type cells?

Intrigued by its localization pattern, we set out to determine the targeting pathway utilized by the Dsc complex. We chose Ubx3 to represent the Dsc complex because: (1) Ubx3 forms a stable complex with the rest of the Dsc components. Antibody-mediated depletion of Ubx3 almost completely depletes the entire Dsc complex from the cell lysate; (2) Ubx3 is the only Dsc component that can be chromosomally tagged with fluorescent proteins without abolishing Dsc function in budding yeast (*Li et al., 2015b*). To begin the investigation, we tagged Ubx3 with mNeonGreen, a green fluorescent protein that is similar to GFP in size, but about two times brighter than GFP (*Shaner et al., 2013*). In the wild type strain, Ubx3-mNeonGreen (hereafter referred to as Ubx3-nG)

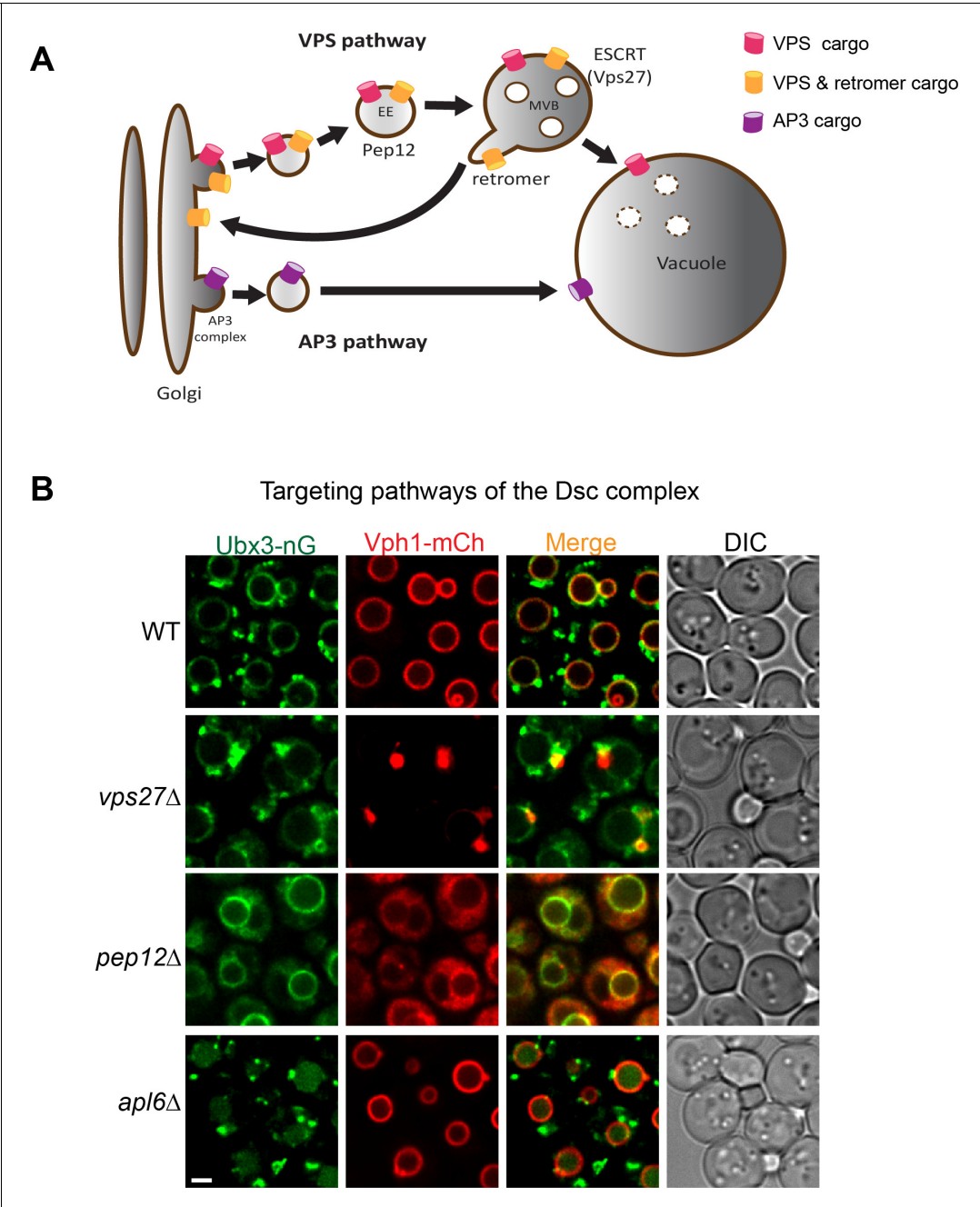

**Figure 1.** Targeting pathways utilized by the Dsc complex. **(A)** A cartoon model showing two conserved pathways (VPS and AP3) for transporting proteins from Golgi to vacuole. Some VPS cargoes can be recycled from the endosome to Golgi by the retromer complex before reaching the vacuole. EE: early endosome, MVB: multivesicular body. **(B)** Localization of Ubx3-nG and Vph1-mCh in WT, *vps27Δ*, *pep12Δ*, and *apl6Δ* strain cells. Scale bar: 2 µm.

DOI: https://doi.org/10.7554/eLife.33116.003

localized to the vacuole membrane and intracellular punctae, consistent with our previous observation (*Figure 1B*)(*Li et al., 2015a*). In contrast, deleting *VPS27*, which encodes an essential component of the ESCRT machinery, led to a partial accumulation of the Ubx3-nG in the class E compartment. A significant amount of Ubx3-nG, however, was still able to reach the vacuole membrane (*Figure 1B*). As a control, Vph1-mCherry (hereafter referred to as Vph1-mCh), a cargo of the VPS pathway, was almost entirely trapped in the class E compartment (*Zhu et al., 2017*).

Furthermore, deleting the gene encoding the endosomal t-SNARE Pep12 blocked the trafficking of Vph1-mCh and led to a 'cytosolic' accumulation of Vph1-mCh as numerous small vesicles (*Figure 1B*). The same *PEP12* deletion, however, only resulted in a partial accumulation of Ubx3-nG in the cytosol, with a significant amount of the protein reaching the vacuole membrane (*Figure 1B*). Together, these results suggest that the Dsc complex might use both VPS and AP3 pathways for its trafficking. However, VPS and AP3 cargoes use different signals for their targeting. How can one protein complex contain both targeting signals? A possible explanation might be that the Dsc complex has two distinct sub-populations. One uses the AP3 pathway, the other uses the VPS pathway.

To test whether a fraction of the Dsc complex uses the AP3 pathway for vacuolar delivery, we deleted *APL6*, which encodes a key component of the AP3 adaptor complex (*Cowles et al., 1997*; *Odorizzi et al., 1998b*). Interestingly, although *APL6* deletion did not change the punctae localization of Ubx3-nG, it led to an accumulation of the Ubx3-nG in the vacuole lumen. The result is different from the reported AP3 pathway cargoes such as ALP and Ypq1, which partially accumulate at the Golgi and partially reach the vacuole membrane via the VPS pathway (*Cowles et al., 1997*; *Li et al., 2015b*), in the absence of the AP3 adaptor complex. This implied that the AP3 Dsc sub-complex, when forced into the VPS pathway, was recognized by an unidentified endosomal protein quality control machinery as an 'abnormal' protein complex and degraded in the vacuole lumen. However, this hypothesis cannot explain the fact that Dsc complex normally exists on the Golgi and endosomes at the steady state, unless the AP3 Dsc sub-population carries a distinct unknown subunit that does not exist in the Golgi/endosome subcomplex.

## Identification of two new components of the Dsc complex

To investigate if there are unknown components in the Dsc complex that determine its trafficking pathways, we chromosomally tagged Ubx3 with the Flag tag at the C-terminus and performed immunoprecipitation (IP) experiments to isolate the Dsc complex. Our previous research has demonstrated that the Dsc complex containing Ubx3-Flag is still able to ubiquitinate a vacuole membrane substrate Cot1 after $Zn^{2+}$ withdrawal (*Li et al., 2015a*), indicating that Ubx3-Flag is still functional. As shown in *Figure 2A–B*, all known components of the budding yeast Dsc complex, including Dsc2, Dsc3, Tul1, and Cdc48, can be co-purified with Ubx3-Flag. Importantly, two new bands at ~27 and ~31 KDa, respectively, co-immunoprecipitated with Ubx3-Flag. Mass spectrometry analysis identified the 27 KDa band as Yir014W (*Figure 2—figure supplement 1A*, hereafter referred to as 014W), and the 31 KDa band as Ypr109W (*Figure 2—figure supplement 1B*, hereafter referred to as 109W), both of which are proteins of unknown function.

To confirm that these two proteins are genuine components of the Dsc complex, we chromosomally labelled them with the Flag tag at the C-termini and performed reciprocal IP experiments. As shown in *Figure 2C and D*, both 014W-Flag and 109W-Flag can pull-down other tested Dsc complex components, including Ubx3, Dsc2, Dsc3, and Tul1, whereas Vph1, an abundant vacuolar membrane protein, cannot be co-immunoprecipitated. Bioinformatics analysis indicated that both proteins have four transmembrane helices, and they share a significant sequence similarity (53%, *Figure 2E* and *Figure 2—figure supplement 1C*) (*Sievers et al., 2011*; *Tsirigos et al., 2015*). We also performed position-specific iterated BLAST (PSI-BLAST) search of sequenced protein databases. Intriguingly, both 014W and 109W showed a low sequence similarity to Dsc4 from several sequenced fungus species, including *Stemphylium lycopersici*, *Escovopsis weberi*, *Tolypocladium ophioglossoides CBS 100239*, and *Ceratocystis fimbriata CBS 114723* (data not shown), suggesting they might be the 'missing' Dsc4 in budding yeast.

We were puzzled by the finding that two similar proteins coexist in the same protein complex. One possible explanation was that 014W and 109W exist in different sub-populations of the Dsc complex, considering that the complex has three distinct sub-cellular locations. Indeed, as shown in *Figure 2F*, in a yeast strain that was co-expressing 014W-HA and 109W-Flag from their chromosomal loci, these two proteins did not co-immunoprecipitate with each other, although each was capable of pulling down all other tested Dsc components, including Ubx3, Dsc2, Dsc3, and Tul1.

Together, we have identified two novel components of the Dsc complex (014W and 109W), which are similar to each other. Each protein is capable of forming a stable complex with the known Dsc components. However, 014W and 109W do not coexist in the same sub-population. These results support the existence of two distinct Dsc subcomplexes within the cell.

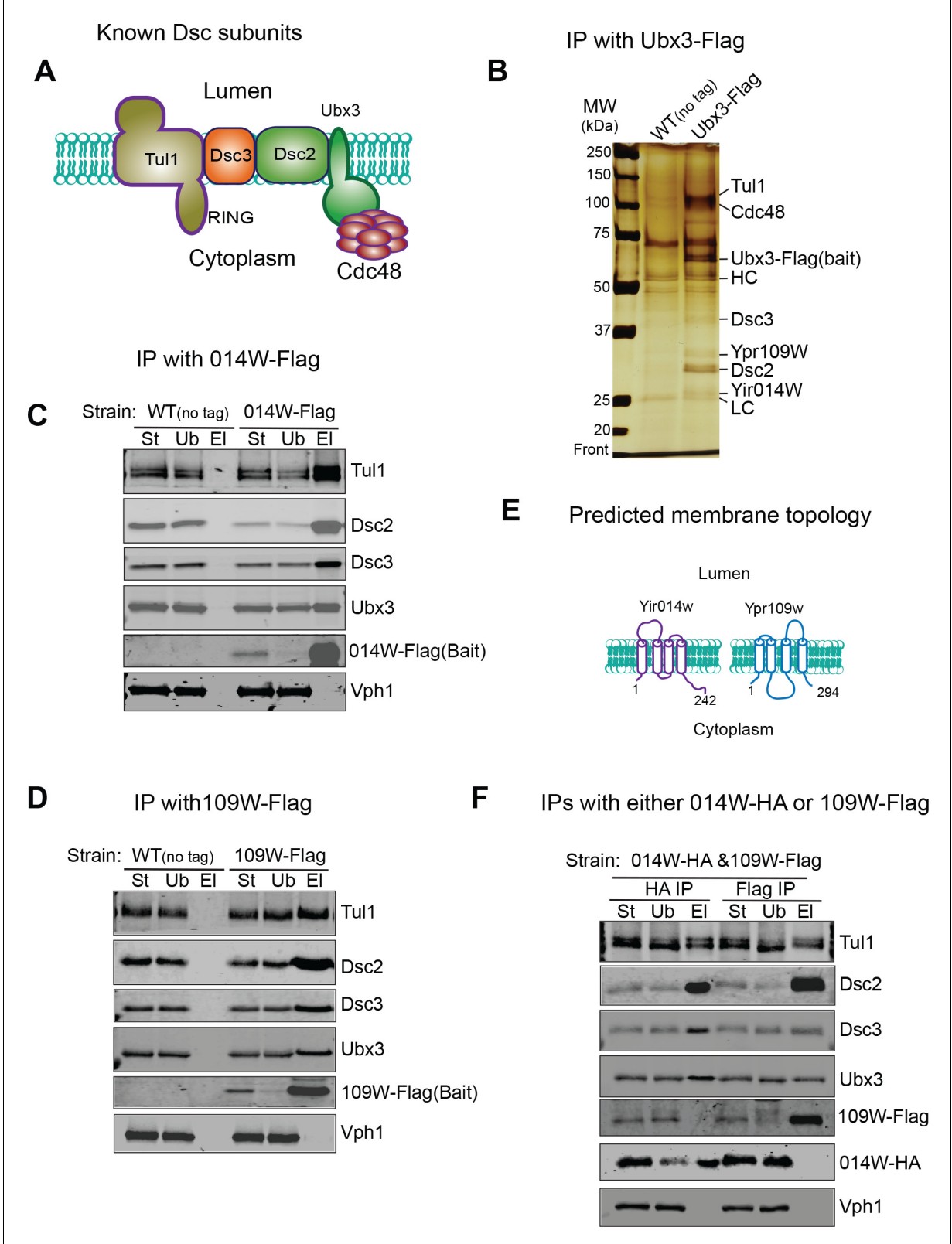

**Figure 2.** Identification of two new Dsc components. (A) A cartoon model showing known Dsc components in budding yeast. (B) A silver staining gel showing that Ypr109W and Yir014W co-immunoprecipitate with Ubx3-Flag, together with other known Dsc components. HC, heavy chain. LC, light chain. (C) 014W-Flag can selectively pull down known Dsc components. (D) 014W-Flag can selectively pull down with known Dsc components. (E)

*Figure 2 continued on next page*

*Figure 2 continued*

Cartoon representations of 014W and 109W. Both proteins are predicted to be tetra-spanning membrane proteins. (**F**) 014W-HA and 109W-Flag do not pull down each other. St: Starting material. Ub: Unbound. El: Elution.

DOI: https://doi.org/10.7554/eLife.33116.004

The following figure supplement is available for figure 2:

**Figure supplement 1.** Identification of two new Dsc subunits by mass spectrometry.

DOI: https://doi.org/10.7554/eLife.33116.005

## 014W and 109W have distinct subcellular locations

To further investigate if 014W and 109W form distinct Dsc subcomplexes, we chromosomally tagged them with mNeonGreen (nG) to directly visualize their sub-cellular localizations. Strikingly, 014W-nG localized exclusively to the FM4-64 labelled vacuole membrane, while 109W-nG only localized to intracellular punctae (~3.7 per cell, n = 110 cells, *Figure 3A–B* and *Figure 3—source data 1*). Further analysis revealed that ~ 37.4% of the 109W punctae co-localized with the Mars-Sec7 labelled trans-Golgi compartment, while ~56.9% of the 109W punctae co-localized with the FM4-64 labelled endosomes (*Figure 3C–D* and *Figure 3—source data 2*). The distinct localizations of 109W and 014W are consistent with a model in which these two proteins form distinct subcomplexes that travel separately to either the Golgi/endosome or the vacuole membrane.

A prediction of our model is that 109W should co-localize with the Ubx3 punctae. However, our attempts to show the co-localization by simultaneously tagging these two proteins with red and green fluorescent proteins proved to be difficult. Mysteriously, tagging either 109W or Ubx3 with different red fluorescent proteins, including mCherry, DsRed, and tagRFP, always resulted in the degradation of the fusion proteins in the vacuole lumen (data not shown). Inspired by the 'knock sideways' technology (*Haruki et al., 2008*; *Robinson et al., 2010*), we developed a new assay, which we named as Rapamycin Induced Co-localization (RICo) assay, to show that 109W co-localizes with the Ubx3-nG punctae. As shown in *Figure 3E*, we fused the FRB moiety to mCherry and expressed the fusion protein under a weak *SSH4* promoter (75 copies per cell). Meanwhile, 109W was chromosomally tagged with FKBP. In the absence of rapamycin, FRB-mCherry appeared to be cytosolic. However, 45 min after the addition of rapamycin, FRB-mCherry was recruited to 109W-FKBP and appeared as punctae that co-localized with Ubx3-nG (*Figure 3E*).

Together, our data suggest that 014W and 109W form distinct Dsc subcomplexes at the vacuole membrane and Golgi/endosomes, respectively. Since both proteins have not been characterized before, we named 014W as Vacuole Localized Dsc protein1 (Vld1) and 109W as Golgi/endosome Localized Dsc protein1 (Gld1).

## Vld1 and Gld1 compete with each other to determine the subcellular localizations of the Dsc complex

Because Vld1 and Gld1 are the only unique components within the vacuole and Golgi/endosome subcomplexes and they share a significant protein sequence similarity, we hypothesized that they may compete with each other to determine the subcellular locations of the Dsc complex. As an initial step, we deleted genes encoding either Vld1 or Gld1 to test if these mutants affect the corresponding Dsc subcellular localizations. As shown in *Figure 4A*, deleting *VLD1* eliminated the vacuole membrane localization of Ubx3-nG, whereas deleting *GLD1* abolished its Golgi and endosome localizations. Intriguingly, deletion of *GLD1* gene also resulted in a partial accumulation of smaller punctae in the endoplasmic reticulum (ER) (*Figure 4A and B*), as indicated by its co-localization with the ER marker DsRed-HDEL (*Figure 4B*), probably because not all Ubx3-nG were assembled into the Dsc complex and the excessive Ubx3-nG were trapped at the ER. This result also suggested that the assembly of the Dsc complex might happen at the ER.

As a direct test of the competition hypothesis, we asked if gradually elevating the expression levels of either Vld1 or Gld1 can recruit more and more Dsc complex to their corresponding subcellular locations. As shown in *Figure 4C*, in a *vld1Δ* strain transformed with an empty vector, Ubx3-nG localized exclusively to the intracellular punctae. However, after expressing Vld1 using its native promoter, a significant amount of Ubx3-nG was recruited to the vacuole membrane from the punctae. Further increasement of the Vld1 expression level with a *GPD* promoter resulted in an exclusive

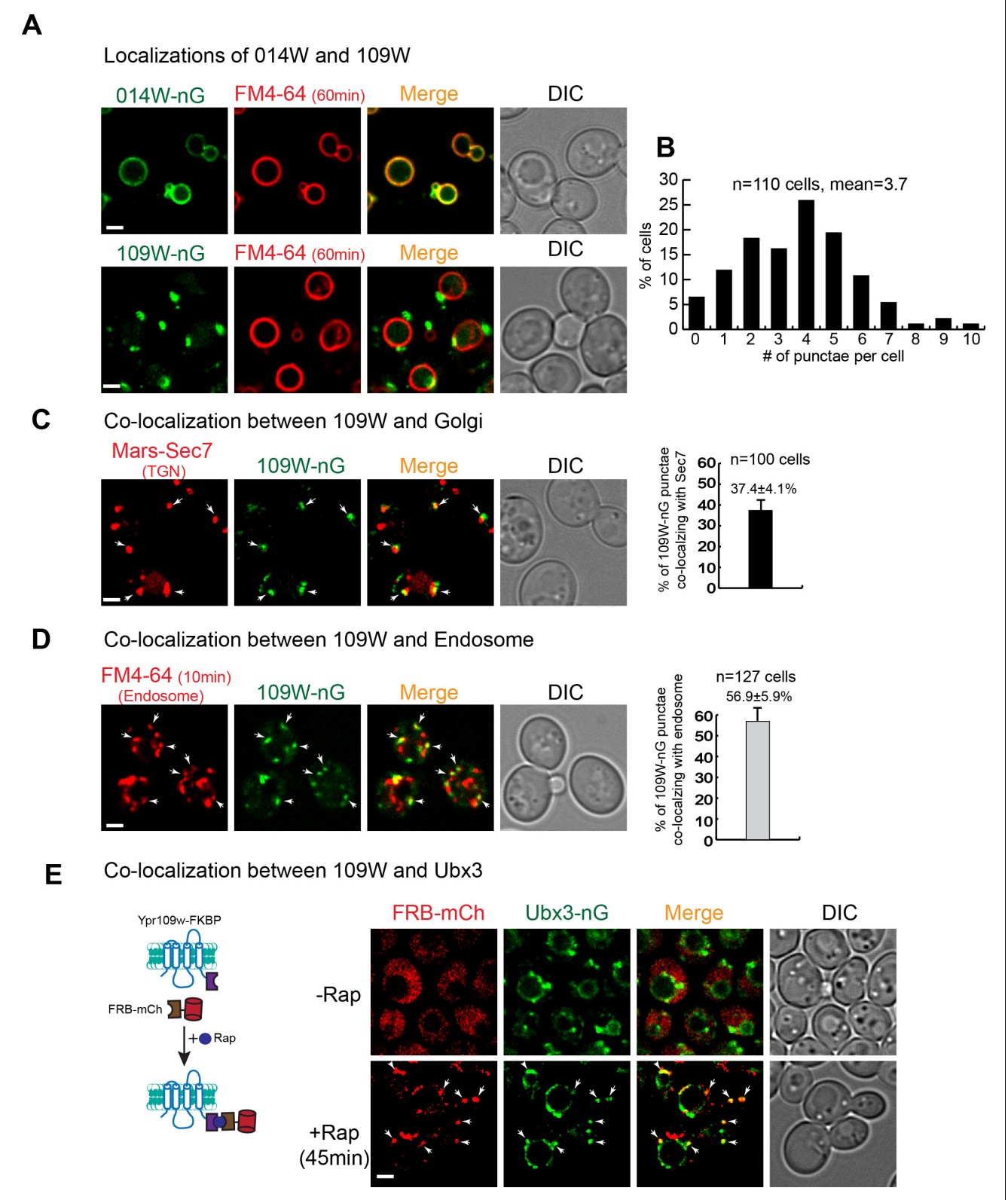

**Figure 3.** 109W and 014W have distinct subcellular localizations.  (**A**) 014W-nG co-localizes with FM4-64 labelled vacuole membrane, whereas 109W-nG localizes to the intracellular punctae. (**B**) Quantification of the number of 109 W-nG punctae. (**C**) Left, 109W-nG partially co-localizes with Mars-Sec7 labelled trans-Golgi (white arrows). Right, quantification of the co-localization. Error bar represents the Standard Error of the Mean (SEM). (**D**) Left, 109W-nG partially co-localizes with FM4-64 labelled endosomes (white arrows). Right, quantification of the co-localization. Error bar represents the

*Figure 3 continued on next page*

*Figure 3 continued*

SEM. (**E**) Co-localization between 109W and Ubx3 using rapamycin induced co-localization (RICo) assay. Left: A cartoon diagram showing the principle of the assay. Right: FRB-mCherry localization before and after rapamycin treatment. White arrows indicate the co-localization. Scale bar: 2 μm.

DOI: https://doi.org/10.7554/eLife.33116.006

The following source data is available for figure 3:

**Source data 1.** The source data for the quantification of 109W-nG punctae in *Figure 3B*.

DOI: https://doi.org/10.7554/eLife.33116.007

**Source data 2.** The source data for the quantification of co-localization in *Figure 3C-D*.

DOI: https://doi.org/10.7554/eLife.33116.008

vacuole membrane localization of Ubx3-nG (*Figure 4C*). Conversely, elevating the Gld1 expression level in a *gld1Δ* strain gradually recruited Ubx3-nG from the vacuole membrane to intracellular punctae (*Figure 4—figure supplement 1A*). We also used immunoprecipitation experiments to verify the imaging results. As shown in *Figure 4—figure supplement 1B*, overexpression of Vld1-HA under the *GPD* promoter can displace Gld1-GFP from the Dsc complex, and vice versa. All together, these experiments provided a direct evidence that Vld1 and Gld1 compete with each other for the Dsc complex.

One prediction of the competition hypothesis is that both vacuole and Golgi/endosome subcomplexes should function independently. As stated above, $Zn^{2+}$ withdrawal leads to the degradation of the vacuolar $Zn^{2+}$ transporter Cot1(*Li et al., 2015a*), whereas amino acid starvation (or rapamycin treatment) triggers the degradation of a Golgi resident protein Yif1(*Dobzinski et al., 2015*). Both processes are initiated by the Dsc complex-mediated protein ubiquitination. As shown in *Figure 5*, deletion of *TUL1*, the RING domain E3 ligase that exists in all Dsc subcomplexes, blocked the degradation of both GFP-Yif1 and Cot1-GFP. GFP-Yif1 was stabilized on the intracellular punctae after 4 hr of amino acid starvation (*Figure 5A*), whereas Cot1-GFP was stabilized on the vacuole membrane (*Figure 5C*). However, deletion of the Golgi/endosome-specific component, Gld1, significantly reduced GFP-Yif1 degradation, whereas Cot1-GFP degradation was unaffected (*Figure 5*). Conversely, deletion of the *VLD1* gene only drastically delayed the degradation of Cot1-GFP without affecting the turnover of GFP-Yif1 (*Figure 5*).

Taken together, we conclude that Vld1 competes with Gld1 to form two functionally independent Dsc subcomplexes that localize to distinct subcellular locations. The Vld1 subcomplex localizes to the vacuole membrane, whereas the Gld1 subcomplex localizes to the Golgi and endosomes. At these locations, they may govern the ubiquitination of distinct organelle membrane proteins.

## Two core complexes that determine Dsc localizations

To determine the minimum subunit requirement for the proper assembly and trafficking of the complex, we took a reductive approach by deleting Dsc subunits and testing if any mutant affects the trafficking of the complex, as indicated by the localizations of Vld1-nG, Gld1-nG, and Ubx3-nG. All three tagged proteins are still functional because they can support the degradation of both Cot1-GFP and GFP-Yif1 (*Figure 6—figure supplement 1*). As shown in *Figure 6A and B*, deletion of either *TUL1* or *DSC3* did not affect the locations of remaining complex components, as demonstrated by Vld1-nG, Gld1-nG, and Ubx3-nG. However, deletion of *DSC2* dramatically changed their localization patterns. After *DSC2* deletion, Vld1-nG, Gld1-nG, and Ubx3-nG were all trapped in smaller punctae that co-localized with the ER marker, DsRed-HDEL (*Figure 6A–C* and *Figure 6—figure supplement 2A–B*). Similarly, deletion of *UBX3* caused the accumulation of Vld1-nG and Gld1-nG as smaller punctae that co-localized with DsRed-HDEL (*Figures 6* and *Figure 6—figure supplement 2A–B*), indicating the importance of Ubx3 in the complex assembly. Lastly, double deletion of *VLD1* and *GLD1* caused the accumulation of Ubx3-nG punctae at the ER (*Figure 6B and C*). Together, our results suggest a model in which the complex assembly may occur at the ER, where Dsc2 and Ubx3 form a core complex with either Vld1 or Gld1 to determine the subcellular localizations. Tul1 and Dsc3 are not part of the core complexes. Consistent with this model, double deletion of *TUL1* and *DSC3* did not affect the proper targeting of Gld1-nG, Vld1-nG, or Ubx3-nG (*Figure 6D*).

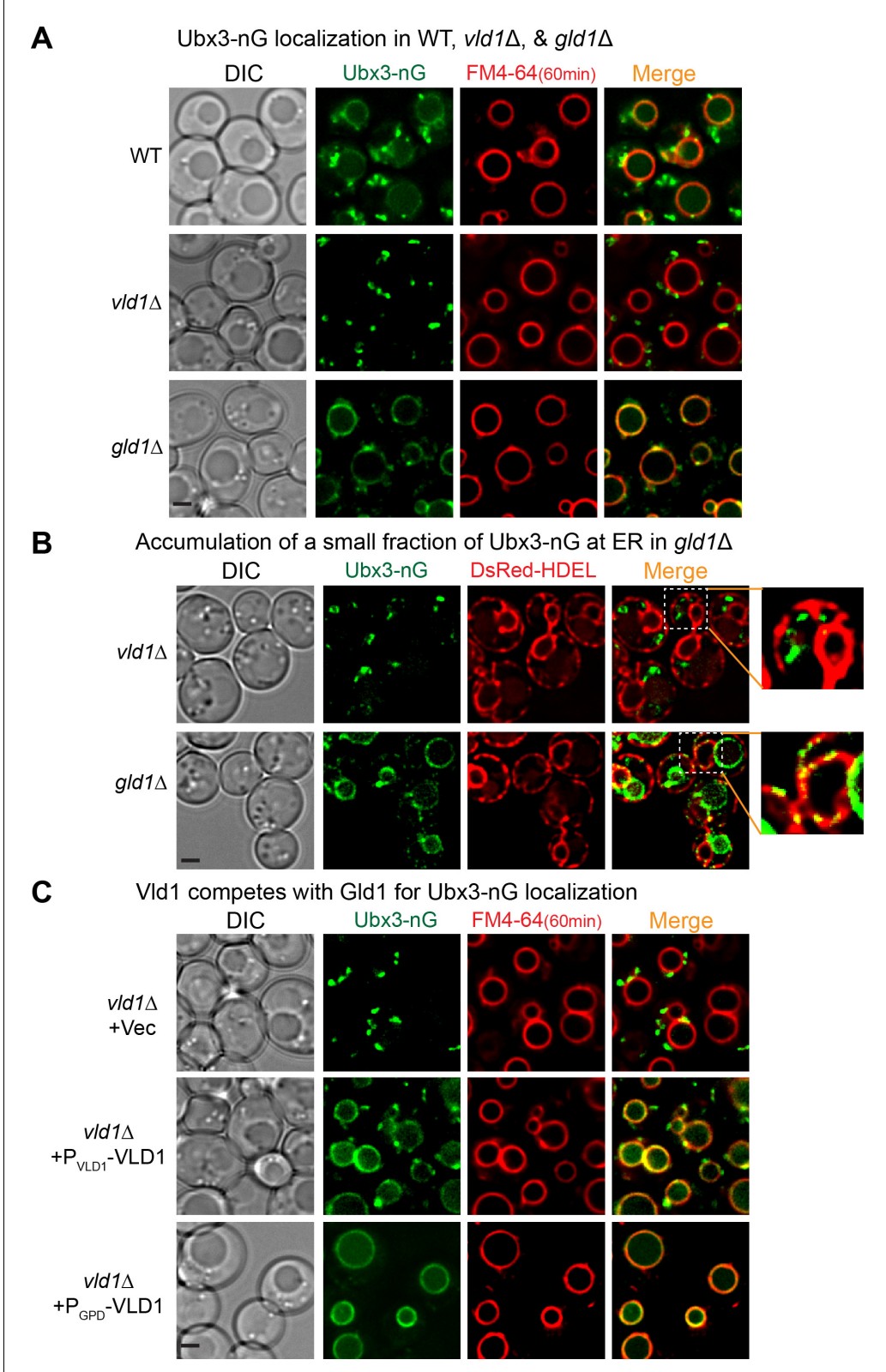

**Figure 4.** Vld1 competes with Gld1 to determine the Dsc subcellular localizations. (**A**) Localization of Ubx3-nG after either *VLD1* or *GLD1* deletion. Note a small amount of punctae are also observed outside the vacuole in *gld1Δ* cells. (**B**) The small Ubx3-nG punctae in the *gld1Δ* cells co-localize with ER marker, DsRed-HDEL, whereas the punctae in *vld1Δ* cells do not co-localize with ER marker. (**C**) A competition assay to show elevating the Vld1 expression level in *vld1Δ* cells gradually diverts Ubx3-nG from punctae to vacuole membrane. Scale bar: 2 μm.

*Figure 4 continued on next page*

*Figure 4 continued*

DOI: https://doi.org/10.7554/eLife.33116.009

The following figure supplement is available for figure 4:

**Figure supplement 1.** Gld1 competes with Vld1 to determine the subcellular localization of the Dsc complex.

DOI: https://doi.org/10.7554/eLife.33116.010

To further verify the core complexes model, we performed a series of IP experiments using various Dsc deletion mutants. As shown in *Figure 6E*, in the absence of Tul1 and Dsc3, Ubx3-Flag was still able to interact with Dsc2, Gld1-GFP, and Vld1-HA. In contrast, either deletion of *DSC2* or the double deletion of both *GLD1* and *VLD1* abolished all the interactions between Ubx3-Flag and the rest of the Dsc subunits. Single deletions of either *GLD1* or *VLD1* did not affect the protein-protein interactions within the counterpart subcomplex (*Figure 6E*). A similar result has been observed by the Espenshade group (*Tong et al., 2014*). Using immunoprecipitation analysis of the Dsc deletion mutants, they concluded that Tul1 and Dsc3 are not essential for the assembly of the complex, whereas Dsc2 and Ubx3 forms a core complex (*Tong et al., 2014*). With our analysis, we now know the core complex should also include either Vld1 or Gld1, and more importantly, the two core complexes determine the subcellular localizations of the Dsc complex.

The observation that *Gld1* and *Vld1* double deletion leads to the ER accumulation of Ubx3-nG enabled us to test if *sp*Dsc4 can support the assembly and trafficking of Dsc complex in budding yeast. As shown in *Figure 6—figure supplement 2C*, overexpression of *sp*Dsc4 under the ADH1 promoter allows Ubx3-nG to traffic out of ER in ~60% of yeast cells. However, the chimeric Dsc complex is unstable and Ubx3-nG appears to accumulate in the vacuolar lumen, presumably due to the low sequence similarity. Nevertheless, this result is consistent with the hypothesis that Gld1 and Vld1 might be the counterpart of Dsc4 in budding yeast.

## Vld1 and Gld1 subcomplexes use two independent trafficking pathways

As shown above in *Figure 1B*, our initial analysis of the Dsc trafficking pathways suggested a confusing model that the complex utilizes both AP3 and VPS pathways for its subcellular localizations. Since the Vld1 subcomplex localizes exclusively to the vacuole membrane, it is reasonable to hypothesize that this subcomplex travels through the AP3 pathway. Indeed, deleting genes encoding either ESCRT machinery, as represented by *vps27Δ*, or the endosomal t-SNARE Pep12, did not affect the vacuole localization of Vld1-nG, although the vacuolar trafficking of Vph1-mCh, a VPS pathway cargo, was abolished in both mutants (*Figure 7A*). In contrast, deleting *APL6* resulted in the accumulation of Vld1-nG on cytosolic punctae, presumably the Golgi and endosome compartment, and the vacuolar degradation of Vld1-nG (*Figure 7A*). The vacuolar degradation of Vld1-nG in the *apl6Δ* strain was consistent with the above mentioned response of Ubx3-nG in the *APL6* deletion strain (*Figure 1B*), suggesting that the entire Vld1 subcomplex, when forced to travel through the VPS pathway, was recognized by the endosomal protein quality control system and turned over in the vacuole lumen. It is very likely that Vld1 was the 'culprit' that caused the ubiquitination and degradation since the remaining subunits are identical between the two subcomplexes.

As stated above, AP3 cargoes normally contain an acidic di-leucine motif for their vacuolar sorting (*Odorizzi et al., 1998b*). Examining the Vld1 protein sequence revealed a putative acidic di-leucine motif ($_{233}\underline{E}ITP\underline{LL}_{238}$) close to the C-terminus of the protein. Fungal sequence alignment indicated that this motif is conserved among orthologues from different fungal species (*Figure 7B*) (*Cliften et al., 2003*; *Kellis et al., 2003*). In order to test if this motif is important for the AP3 pathway trafficking, we mutated the sequence to $_{233}\underline{A}ITP\underline{AA}_{238}$. However, these changes caused an instability of Vld1-nG, and nearly all of it was degraded by an unknown mechanism (data not shown). As an alternative, we deleted the last six amino acids of Vld1 ($_{237}\underline{LL}NIAE_{242}$), including the di-leucine ($vld1^{\Delta6AA}$-nG). This deletion caused a partial punctate (Golgi and endosomes) accumulation and partial vacuolar degradation of $vld1^{\Delta6AA}$-nG in wildtype cells (*Figure 7C*), a phenotype similar to that of Vld1-nG in the *apl6Δ* strain (*Figure 7A*). These results indicated the importance of the di-leucine motif. Further deletion of the *PEP12* gene, which eliminates the VPS pathway, completely blocked the $vld1^{\Delta6AA}$-nG trafficking and the protein appeared as numerous tiny 'cytosolic' punctae outside

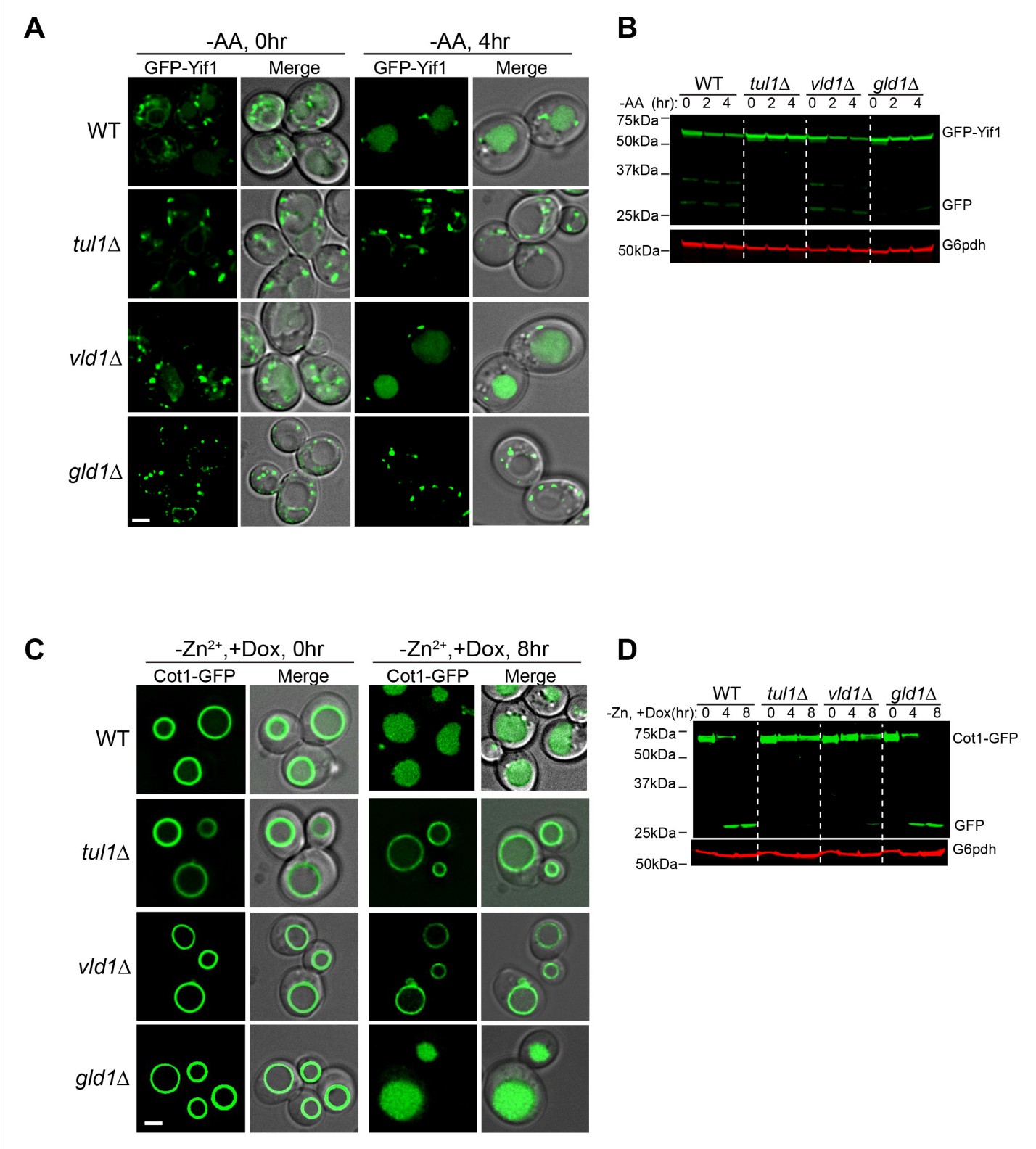

**Figure 5.** The Golgi/endosome and vacuole subcomplexes are functionally independent. (**A**) Subcellular localizations of GFP-Yif1 before (0 hr) and after (4 hr) amino acid starvation. (**B**) Western blot analysis of the GFP-Yif1 starvation assay. 1 $OD_{600}$ cells were loaded in each lane. (**C**) Subcellular localizations of Cot1-GFP before (0 hr) and after (8 hr) $Zn^{2+}$ withdrawal. (**D**) Western blot analysis of the Cot1-GFP degradation assay. Same volume of cells was loaded, with 0.25 $OD_{600}$ cells loaded at 0 hr. Scale bar: 2 μm.

DOI: https://doi.org/10.7554/eLife.33116.011

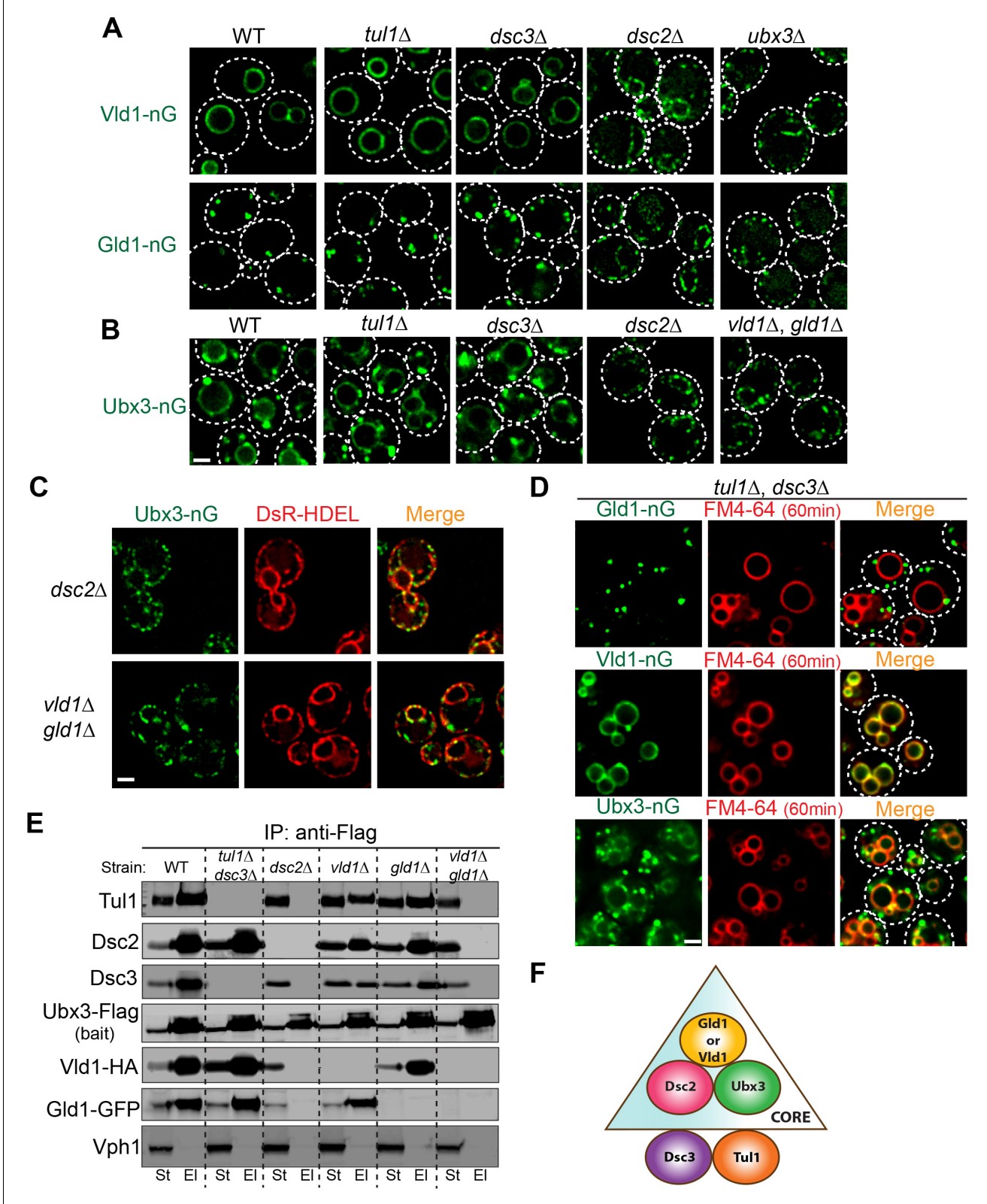

**Figure 6.** Two core complexes that determine the subcellular localizations. (**A**) Deletion analysis for Vld1-nG and Gld1-nG localizations. (**B**) Deletion analysis for Ubx3-nG localization. (**C**) Co-localization between the Ubx3-nG punctae and ER marker DsRed-HDEL in either *dsc2Δ* or *vld1Δ gld1Δ* strains. (**D**) Double deletion of *TUL1* and *DSC3* did not change the localization patterns of Vld1-nG, Gld1-nG, or Ubx3-nG. (**E**) Western blot analysis of the Dsc

*Figure 6 continued on next page*

*Figure 6 continued*

complex assembly after deleting indicated Dsc components. St: Starting material, El: Elution. (F) A model to summarize the core complex concept. White dashed lines indicate the periphery of yeast cells. Scale bar: 2 µm.

DOI: https://doi.org/10.7554/eLife.33116.012

The following figure supplements are available for figure 6:

**Figure supplement 1.** Ubx3, Vld1 and Gld1 still support the degradation of Cot1-GFP and GFP-Yif1 after neonGreen tagging.

DOI: https://doi.org/10.7554/eLife.33116.013

**Figure supplement 2.** Both Vld1-nG and Gld1-nG are trapped in the ER after deleting either *DSC2* or *UBX3*.

DOI: https://doi.org/10.7554/eLife.33116.014

the vacuole (*Figure 7C*). In contrast, the trafficking of an AP3 pathway cargo, mCh-ALP, was not affected by the *PEP12* deletion. We also tested whether the acidic $E_{233}$ residue is important for the Vld1 trafficking. As shown in *Figure 7D*, a single $E_{233}$ to $A_{233}$ mutation (*vld1*$^{E233A}$-nG) was sufficient to cause trafficking defects similar to *vld1*$^{\Delta6AA}$–nG. All together, these results strongly indicate that the putative acidic di-leucine motif is important for the trafficking of Vld1 through the AP3 pathway.

The observation that Gld1 co-localizes with FM4-64 labelled endosomes (*Figure 3C and D*) supports the hypothesis that the protein travels through the VPS pathway. To confirm this, we tested if the ESCRT or *PEP12* deletion mutants can affect the trafficking of Gld1. As shown in *Figure 7E*, a fraction of the Gld1-nG punctae co-localized with the Vph1-mCherry labelled class E compartment in *vps27Δ* strain. The Golgi population of Gld1-nG, however, did not accumulate in the class E compartment. In the *pep12Δ* strain, Gld1-nG trafficking was also disrupted and the protein appeared as 'cytosolic', with very few punctae left. This phenotype was also similar to that of the VPS pathway cargo Vph1-mCherry (*Figure 7E*). To further verify that Gld1 traffics through the VPS pathway, we tested if Gld1 is cycled between the Golgi and endosomes by the retromer, a sorting apparatus that functions on the endosomes to prevent important endosomal proteins from mis-targeting to the vacuole membrane (*Burd and Cullen, 2014*; *Seaman et al., 1997*). As shown in *Figure 7F*, deleting *VPS35*, an essential subunit in the retromer complex, resulted in the mis-localization of Gld1-nG to the vacuole membrane.

Taken together, we conclude that the Dsc subcomplexes use two independent trafficking pathways for their subcellular localizations. The Vld1-containing subcomplex traffics through the AP3 pathway to reach the vacuole membrane, whereas the Gld1-containing subcomplex travels through the VPS pathway and is cycled between Golgi and endosomes by the retromer machinery (*Figure 7G*).

## Discussion

### One RING to 'rule' three organelles

Protein ubiquitination and degradation is an essential process to rapidly down-regulate protein levels and remove unfolded/damaged proteins in eukaryotic cells. How to recognize so many different substrates in a regulated fashion using a limited number of E3 ligases is a fundamental challenge faced by all eukaryotic cells. For example, the human proteome contains ~19,000–20,000 different proteins (*Kim et al., 2014*; *Wilhelm et al., 2014*), but only has ~640 E3 ligases (*Morreale and Walden, 2016*). This challenge is further complicated by the presence of different organelles that divide the cell into numerous compartments, which requires the proper targeting of E3 ligases to different organelle surfaces. The human proteome has 6,000–7,000 predicted transmembrane proteins. However, there are only ~50 predicted transmembrane E3 ligases (*Lussier et al., 2012*; *Nakamura, 2011*). With so few transmembrane E3 ligases, how does the cell achieve the regulated ubiquitination for so many membrane proteins?

During evolution, eukaryotic cells have developed several ingenious ways to expand the substrate repertoire of existing E3 ligases. At the protein level, one strategy is through the expansion of the interchangeable F-box proteins in the SCF E3 ligase complex. F-box is an approximately 50 amino acids motif that mediates protein-protein interaction (*Kipreos and Pagano, 2000*). As part of the SCF E3 ligase complex, F-box proteins are responsible for recognizing specific substrates. Many SCF complexes only differ in their F-box protein subunits. Therefore, by expanding the F-box protein

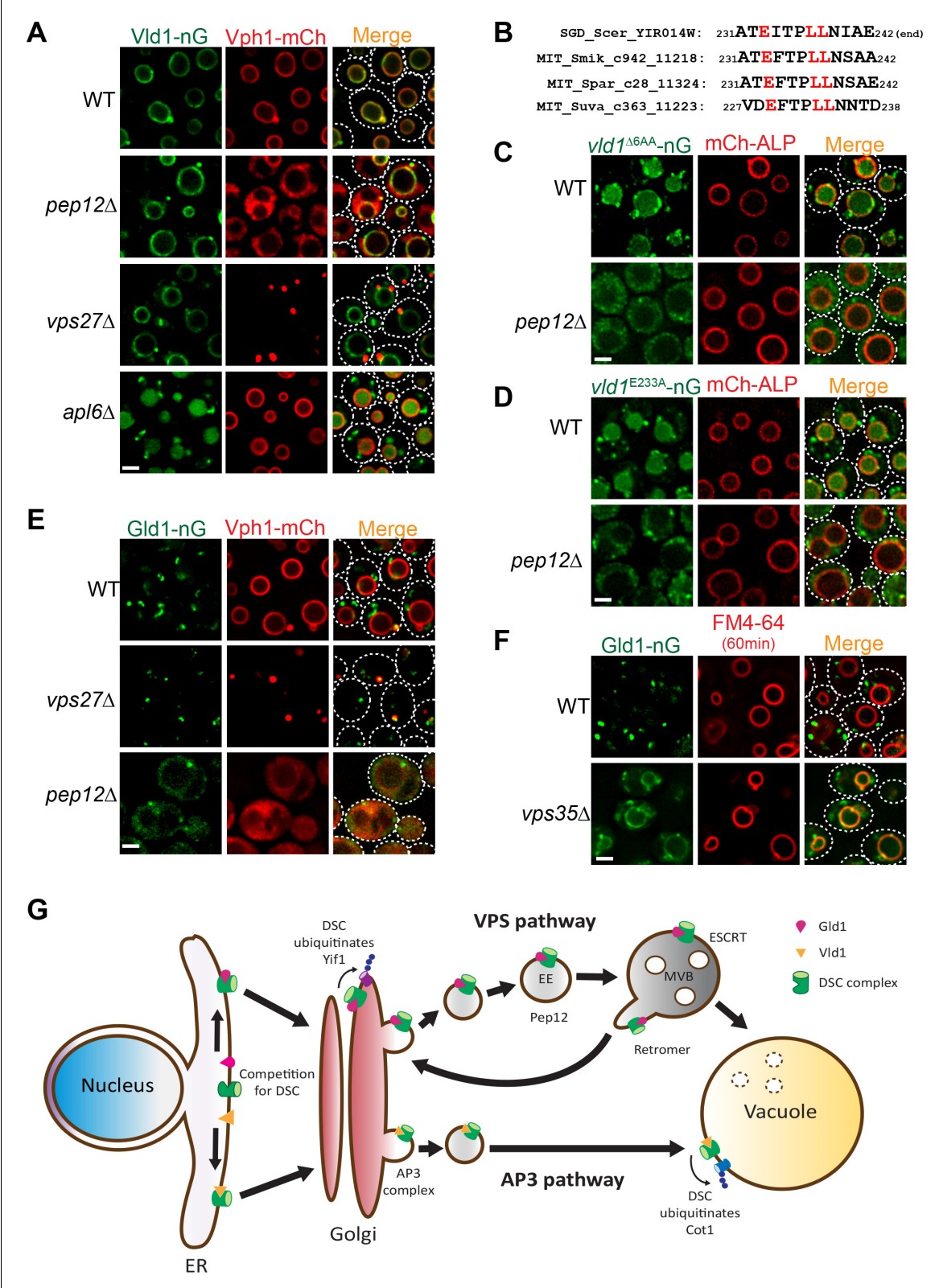

**Figure 7.** Vld1 and Gld1 subcomplexes travel through the AP3 and VPS pathways, respectively. (**A**) Subcellular localizations of Vld1-nG and Vph1-mCh in WT, *pep12Δ*, *vps27Δ*, and *apl6Δ* cells. (**B**) Vld1 contains a conserved acidic di-leucine motif at its C-terminus. (**C**) Subcellular localizations of *vld1*[Δ6AA]-nG (last 6 amino acids of Vld1 deleted) in WT cell and *pep12Δ* cells. (**D**) A single E233 to A mutation caused the trafficking defects of *vld1*[E233A]-nG in both WT and *pep12Δ* cells. (**E**) Subcellular localizations of Gld1-nG and Vph1-mCh in WT, *vps27Δ*, and *pep12Δ* cells. (**F**) Subcellular localizations of
*Figure 7 continued on next page*

*Figure 7 continued*

Gld1-nG in WT and *vps35Δ* cells. (G) A model summarizing the key findings of this study. White dashed lines indicate the periphery of yeast cells. Scale bar: 2 μm.

DOI: https://doi.org/10.7554/eLife.33116.015

family, the cell can recognize different substrates using a similar SCF complex. For example, although budding yeast has only 11 F-box proteins, *C. elegans* has more than 300 F-box proteins (*Kipreos and Pagano, 2000*), and *A. thaliana* has over 1,300 F-box proteins (*Hua et al., 2011*). At the organelle level, another strategy is to use adaptor proteins to recruit cytosolic E3 ligases to different organelle surface. For example, in budding yeast, the NEDD4 family E3 ligase Rsp5 can be recruited to the plasma membrane (*Lin et al., 2008*; *MacGurn et al., 2012*), Golgi (*Hettema et al., 2004*), endosomes (*Léon et al., 2008*), and vacuole (*Li et al., 2015b*) by different PY motif containing proteins (*Wang et al., 2001*). The PY motif interacts with the WW motifs of Rsp5 to recruit the ligase. A similar recruitment mechanism has been observed for the human NEDD4 family members. The human NEDD4-2 can be recruited to the plasma membrane by the PY motif within the surface epithelial sodium channel (ENaC) protein in order to down-regulate ENaC (*Fotia et al., 2003*).

In this study, we report a novel mechanism to directly target a membrane-residing E3 ligase complex to different organelles. Through the characterization of targeting pathways of the budding yeast Dsc complex, we unexpectedly uncovered that the ER can assemble two distinct Dsc subcomplexes using Gld1 and Vld1. Gld1 and Vld1 then guide the Dsc complex through two independent trafficking pathways (VPS and AP3) to reach Golgi/endosomes and lysosomes, respectively. This mechanism allows the cell to achieve protein regulation and probably quality control at three distinct organelles, namely Golgi, endosomes, and vacuole, using just one RING domain E3 ligase Tul1 (*Figure 7G*). Because both VPS and AP3 pathways are conserved from yeast to human (*Bonifacino and Traub, 2003*), it is reasonable to speculate that human cells might use a similar strategy to expand the substrate repertoire of their membrane E3 ligases. Consistent with this hypothesis, several human RING domain E3 ligases have been identified in the endomembrane trafficking pathways, such as March2, March3, March11, RNF152, and RNF167 (*Deng et al., 2015*; *Lussier et al., 2012*; *Nakamura, 2011*; *Nakamura et al., 2005*). It remains to be verified whether there is a similar multi-localization mechanism for these E3 ligases.

## Protein quality control on the endosomes

Endosomes are intermediate transporting organelles between the Golgi (or plasma membrane for endocytosis) and lysosomes. On the endosome, ubiquitinated membrane cargoes are sorted and internalized into the lumen as intraluminal vesicles by the ESCRT machinery, whereas cargo receptors such as LDL receptor in human and Vps10 in yeast will be recycled back to either plasma membrane or Golgi (*Goldstein and Brown, 2009*; *Seaman et al., 1997*). As stated above, many studies have reported that endosomes also contain E3 ligase systems in yeast, plants, and metazoans (*Léon et al., 2008*; *Nakamura, 2011*; *Tian et al., 2015*; *Voiniciuc et al., 2013*). However, it remains to be addressed as to the functions of these endosomal E3 ligases. One reasonable hypothesis is to counteract the activity of endosomal deubiquitinases (Dubs) (*Kee et al., 2005*; *Kee et al., 2006*; *Léon et al., 2008*). During the endomembrane trafficking, ubiquitinated cargoes can be deubiquitinated by the Dubs and later recycled by the retromer. The presence of endosomal E3 ligases enables the cell to have the capability of determining whether a cargo needs to be degraded or recycled at the endosome stage (*Léon et al., 2008*).

In this study, we made a surprising observation that the Vld1-containing Dsc subcomplex, when forced into the VPS pathway, was recognized by an unidentified endosomal protein quality control (EQC) system and degraded inside the vacuole lumen (*Figures 1B* and *7A*). This observation indicated that endosomal E3 ligases may indeed have the capability of recognizing mislocalized proteins and removing them, although the identity of the E3 ligase still needs to be addressed. Strikingly, the only difference between the two subcomplexes were the Vld1 and Gld1 subunits, which are homologous to each other. Yet, the EQC system was able to recognize Vld1 and selectively ubiquitinate the protein. Considering endosomes are constantly receiving exogenous proteins delivered by the transporting vesicles, it is stunning to realize that endosomes can tell the difference between transporting

cargoes and mislocalized membrane proteins. Apparently, future investigations, such as identifying the E3 ligase and expanding the substrate list of the EQC system, are needed to address this striking quality control mechanism.

# Materials and methods

**Key resources table**

| Reagent type (species) or resource | Designation | Source or reference | Identifiers | Additional information |
|---|---|---|---|---|
| Strain, strain background (54 yeast strains) | please find in *Supplementary file 1* | | | |
| Transfected construct (11 yeast constructs) | please find in *Supplementary file 1* | | | |
| Antibody | rabbit anti-G6PDH | Sigma-Aldrich | A9521 | |
| Antibody | mouse anti-GFP | Santa Cruz Biotechnology, Inc. | SC9996 | |
| Antibody | mouse anti-Flag | Sigma-Aldrich | F7425 | |
| Antibody | rabbit anti-HA | Life technologies | 715500 | |
| Antibody | mouse anti-HA | Sigma-Aldrich | 12CA5 | |
| Antibody | mouse anti-Vph1 | Invitrogen | 10D7 | |
| Antibody | rabbit anti-GFP | Torrey Pines Biolabs | TP401 | |
| Antibody | M2 anti-FLAG resin | Sigma-Aldrich | A2426 | |
| Antibody | anti-HA resin | Sigma-Aldrich | E6779 | |
| Peptide | 3xFlag peptide | Sigma-Aldrich | F3290 | |
| Chemical compound | Rapamycin | LC Laboratories | ASW-135 | |
| Chemical compound | Doxycycline | Fisher BioReagents | 162209 | |
| Chemical compound | zymolyase | Amsbio | 120493–1 | |
| Chemical compound | FM4-64 | Invitrogen | T3166 | |
| Chemical compound | Protease Inhibitor Cocktail | Roche | 21169500 | |

## Yeast strains, Plasmids, Media, and Growth Conditions

All yeast strains and plasmids used in this study are listed in *Supplementary file 1*. Both Difco YPD broth and Difco Yeast Nitrogen Base (YNB) w/o amino acids were purchased from Thermo Fisher Scientific. All yeast strains were grown at 26°C in either YPD or YNB media before further analysis.

## Mass spectrometry analysis

The MS analysis was performed by the Mass Spectrometry and Metabolomics Core at the Michigan State University. Essentially, eluted samples from IP experiments were separated by SDS-PAGE and stained with SYPRO Ruby protein gel stain (S12000, Invitrogen). Two bands at 27 KDa and 31 KDa were excised and in-gel digested with trypsin. Then, digested peptides were extracted and eluted peptides were sprayed into a ThermoFisher Q-Exactive mass spectrometer using a FlexSpray spray ion source. Survey scans were taken in the Orbi trap (35000 resolution, determined at m/z 200) and the top ten ions in each survey scan are then subjected to automatic higher energy collision induced dissociation (HCD) with fragment spectra acquired at 17,500 resolution. The resulting MS/MS spectra were converted to peak lists using Mascot Distiller, v2.6.1 and searched against a database of all protein sequences available from SwissProt using the Mascot searching algorithm, v 2.6.0. The Mascot output was then analyzed using Scaffold, v4.7.5 to probabilistically validate protein identifications. Assignments validated using the Scaffold 1%FDR confidence filter were considered true.

## Cot1-GFP degradation assay

The $Zn^{2+}$ minus YNB media was prepared according to the methods of Li et al(*Li et al., 2015a*). For the Cot1-GFP degradation assay, yeast cells were grown in YNB media to mid-log phase ($OD_{600}$: 0.4~0.8), after 20 min of pre-incubation with 2 µg/ml doxycycline, the cells were collected at 2500 g

for 5 min. After two times of washing with water, the cells were resuspended in $Zn^{2+}$ minus YNB media that contained 2 ug/ml doxycycline and incubated at 26°C for an appropriate amount of time (typically, 6–8 hr) before further analysis. In our previous experiments at Cornell University, normally 4–6 hr was enough for a complete degradation of Cot1-GFP in WT. After moving to University of Michigan, we found that the degradation of Cot1-GFP was relatively slower, probably due to a higher level of residual $Zn^{2+}$ in the MilliQ water system, so we have to extend the $Zn^{2+}$ withdrawal treatment to 6–8 hr to get a complete degradation of Cot1-GFP.

## GFP-Yif1 degradation assay

For GFP-Yif1 degradation assay, yeast cells were grown in YNB media to mid-log phase ($OD_{600}$: 0.4~0.8), then collected at 2500 g for 5 min. After two times of washing with water, the cells were resuspended in nitrogen starvation medium (YNB lacking amino acids and ammonium sulfate, with 2% glucose) and incubated at 26°C for an appropriate amount of time (typically, 2–4 hr) before being collected for further analysis.

## Microscopy and image processing

Microscopy was performed with a DeltaVision Elite system (GE Healthcare Life Sciences), equipped with an Olympus IX-71 inverted microscope, a sCMOS camera, a 100X/1.4 Oil Super-Plan APO objective, and a DeltaVision Elite Standard Filter Set with the FITC filter (Excitation:475/28, Emission: 525/48) for mNeonGreen and the TRITC filter (Excitation:542/27, Emission: 594/45) for mCherry, DsRed, and FM4-64. Yeast cells, except for those transformed with the pRS416-DsRed-HDEL plasmid, were grown in YPD at 26°C overnight. Yeast cells transformed with the pRS416-DsRed-HDEL plasmid were grown in YNB minus uracil at 26°C overnight. Before imaging, yeast cells were briefly washed with water and immediately imaged in milliQ water at room temperature. Image acquisition, deconvolution, and maximum projection analysis were performed with the program softWoRx. The image cropping and adjustment were performed using the ImageJ software (National Institutes of Health).

## RICo assay

A rapamycin resistant strain (SEY6210.1, tor1-1, fpr1Δ::NAT)(*Zhu et al., 2017*) was used to develop the RICo assay. First, Ubx3 and Gld1 were chromosomally tagged with mNeonGreen and 2xFKBP, respectively. Then, pRS305-pSSH4-FRB-mCherry plasmid was integrated into the yeast genome for stably expressing FRB-mCherry under the SSH4 promoter. Yeast cells were grown in YPD media to $OD_{600}$ ~3, before being incubated with 1 μg/ml rapamycin at 26°C for 45 min, then collected for imaging.

## FM4-64 staining

Yeast cells were grown overnight in YPD to late log phase. 1–1.5 ml of culture was collected, washed once with 1 ml YNB complete media, and resuspended with 100 ul of YNB complete media. Yeast cells were then labeled with FM4-64 (T3166, Invitrogen, 10 ug/ml final concentration) for 10 min at room temperature in the dark (*Vida and Emr, 1995*). For the endosome staining, cells were immediately washed with 1 ml ice cold YNB complete media to remove the FM4-64 and kept on ice to stop the membrane trafficking. For vacuole membrane staining, 1 ml room temperature YNB complete media was added to the 10 min FM4-64 incubating cells and the incubation was continued for another 50 min in the dark to ensure the vacuole membrane staining. The cells were then collected, and washed with 1 ml ice cold YNB complete media, and kept on ice. Cells were resuspended in milliQ water and imaged by fluorescence microscopy.

## Immunoprecipitation (IP) Assay

The immunoprecipitation assay was adapted from Li et al (*Li et al., 2015a*), with some modifications. Essentially, 1 liter of yeast culture ($OD_{600}$ ~1.5) was harvested by spinning at 4,000 g for 10 min, resuspended in 50 ml weakening buffer (100 mM Tris-HCl, pH 8.8, and 10 mM DTT) and incubated at room temperature for 10 min to weaken the cell wall. The cells were then resuspended with 25 ml spheroplasting media (2% glucose, 1 × amino acids, 1M Sorbitol, 20 mM Tris-HCl, pH 7.5, in YNB) containing 100 μl of 10 mg/ml Zymolyase 100T(120493–1, Amsbio), and incubated at 30°C for 30

min with gentle rocking. After washing once with 20 ml spheroplasting media, the cells were resuspended with 20 ml lysis buffer (20 mM HEPES, pH 7.2, 50 mM KOAc, 10 mM EDTA, 200 mM Sorbitol, with Protease Inhibitor Cocktail (21169500, Roche, Switzerland) and ruptured on ice by 20 strokes in a Dounce homogenizer. The membrane fraction was collected by a 10 min 13,000 g spin at 4°C before being resuspended in 1 ml IP buffer (50 mM Hepes-KOH, pH 6.8, 150 mM KOAc, 2 mM MgOAc, 1 mM CaCl$_2$, 15% glycerol) supplemented with protease inhibitors. Then, the resuspended membrane was dissolved in 10 ml IP buffer containing 1% Triton X-100 at 4°C for 30 min, with gentle rocking. Insoluble material was removed by spinning at 14000 rpm (Sorval SS-34 rotor) for 10 min. The resulting lysate was incubated with 70 µl either M2 anti-FLAG resin (A2426, Sigma-Aldrich, St Louis, MO) or anti-HA resin (E6779, Sigma-Aldrich) for 3 hr at 4°C, with gentle rocking. The resin was then washed six times with 0.1% Triton X-100 in IP buffer. For anti-FLAG resin, bound proteins were eluted with 200 µl 3xFLAG peptide (F3290, Sigma-Aldrich, 100 µg/ml, dissolved in IP buffer containing 0.1% Triton X-100). For anti-HA, the resin was incubated with 2 × SDS PAGE sample buffer (150 mM Tris, pH 6.8, 2% SDS, 100 mM DTT and bromophenol blue) at 42°C for 5 min to dissociate bound proteins. The resulting eluates were then analyzed by either Western blotting or silver staining.

## Sample preparation for western blotting and antibodies

Total cell lysates were prepared from 7 OD$_{600}$ cultures by incubating on ice for 1 hr in 10% TCA. After washing once with 0.1% TCA, samples were bead-beated for 5 min in 2 × urea buffer (150 mM Tris, pH 6.8, 6 M urea, 6% SDS) and incubated 5 min at 65°C. After addition of 2 × sample buffer (150 mM Tris, pH 6.8, 2% SDS, 100 mM DTT and bromophenol blue), samples were bead-beated again for 5 min and heated for another 5 min at 65°C. Samples were then centrifuged at 21,000 g for 5 min and the supernatant was collected. The samples were separated by 10% polyacrylamide gels and transferred to nitrocellulose membranes for western blotting analysis. Antibodies used in this study were G6PDH (A9521; Sigma-Aldrich), mouse anti-GFP (SC9996; Santa Cruz Biotechnology, Inc. Santa Cruz, CA), rabbit anti-GFP (TP401; Torrey Pines Biolabs, Secaucus, NJ ), Flag (F7425; Sigma-Aldrich), rabbit anti-HA (715500, Life technologies, Camarillo, CA), mouse anti-HA (12CA5; Sigma-Aldrich) and Vph1 (10D7, Invitrogen, Carlsbad, CA). Antibodies against Dsc2, Dsc3, Ubx3, and Tul1 were generous gifts from P. Espenshade (Johns Hopkins University, Baltimore, MD).

## Acknowledgements

We thank the undergraduate members of the Li laboratory, including A Hamlin, G Chu, A Kappagantu, P Bulinski, and Y Liu for their technical support. We are also grateful to our colleagues in the Protein Folding and Disease Hub and the MCDB department at the University of Michigan, especially L Weisman, M Duncan, H Xu, R Stockbridge, A Chang, and Y Wang for the helpful discussion and critical reading of the manuscript. M Li is indebted to S Emr for his continued and unreserved support and mentorship. This research is supported by a startup fund and the MCubed fund from the University of Michigan to M Li.

## Additional information

### Funding

| Funder | Grant reference number | Author |
| --- | --- | --- |
| University of Michigan | Startup fund | Ming Li |
| University of Michigan | MCubed fund | Ming Li |

The funders had no role in study design, data collection and interpretation, or the decision to submit the work for publication.

### Author contributions

Xi Yang, Conceptualization, Data curation, Formal analysis, Validation, Investigation, Visualization, Methodology, Writing—original draft, Writing—review and editing; Felichi Mae Arines, Weichao

Zhang, Formal analysis, Validation, Investigation, Writing—review and editing; Ming Li, Conceptualization, Resources, Supervision, Funding acquisition, Investigation, Writing—original draft, Project administration, Writing—review and editing

**Author ORCIDs**
Xi Yang (iD) http://orcid.org/0000-0003-2741-2294
Felichi Mae Arines (iD) http://orcid.org/0000-0002-5770-3116
Weichao Zhang (iD) http://orcid.org/0000-0002-0835-890X
Ming Li (iD) http://orcid.org/0000-0002-1247-2377

**Decision letter and Author response**
Decision letter https://doi.org/10.7554/eLife.33116.019
Author response https://doi.org/10.7554/eLife.33116.020

## Additional files

**Supplementary files**
• Supplementary file 1. Yeast strains and plasmids used in this study.
DOI: https://doi.org/10.7554/eLife.33116.016

• Transparent reporting form
DOI: https://doi.org/10.7554/eLife.33116.017

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
