## [Decision Letter]

Thank you for submitting your article "Sorting of a Multi-subunit Ubiquitin Ligase Complex in the Endolysosome System" for consideration by *eLife*. Your article has been favorably evaluated by Randy Schekman (Senior Editor) and three reviewers, one of whom, Christopher G Burd (Reviewer #1), is a member of our Board of Reviewing Editors.

The reviewers have discussed the reviews with one another and the Reviewing Editor has drafted this decision to help you prepare a revised submission.

Summary:

The DSC complex is an integral membrane E3 ubiquitin ligase that is broadly localized to organelles of the secretory (Golgi) and endo-vacuolar system of yeast, *Saccharomyces cerevisiae*. This work shows that two forms of the DSC complex exist that differ by inclusion of one of two different subunits. One subunit targets DSC via the AP3 sorting pathway to the vacuole, and the other subunit targets DSC to the Golgi and endosome compartments where each form acts on different substrates.

Essential revisions:

1) gDSC and vDSC should not be presented as an "endosomal protein quality control" system. The presentation of the study highlights the role of the DSC complex as a component of an "endosomal protein quality control" system, using analogy to the ERAD pathway that recognizes and degrades misfolded proteins at the ER. This is in line with the original description of Tul1 activity, but this is the only publication that implicates DSC in quality control per se. Protein "quality" in this sense is not addressed in this study, nor is it clear what aspect(s) of DSC substrates the authors are proposing is recognized. The prevailing view of the field is that the ubiquitin ligases (and their adapters) that operate on nutrient and ion transporters in the endo-vacuolar system recognize open versus closed conformations; these are well ordered, physiological conformations, not misfolded states. The reviewers suggest that you present your study in terms of "regulated ubiquitination" or a related term. The existence of an "endosomal protein quality control" system that harnesses the DSC complexes could be incorporated appropriately in the Discussion section of the manuscript.

2) The statement that Vld1 and Gld1 "govern the ubiquitination of different organelle membrane proteins independently" should only be a suggestion, as it is not experimentally addressed.

3) The figure legends need to be rewritten to describe strains used and experimental conditions. Currently, it is written more like a Results section.

---

## [Author Response]

Essential revisions:1) gDSC and vDSC should not be presented as an "endosomal protein quality control" system. The presentation of the study highlights the role of the DSC complex as a component of an "endosomal protein quality control" system, using analogy to the ERAD pathway that recognizes and degrades misfolded proteins at the ER. This is in line with the original description of Tul1 activity, but this is the only publication that implicates DSC in quality control per se. Protein "quality" in this sense is not addressed in this study, nor is it clear what aspect(s) of DSC substrates the authors are proposing is recognized. The prevailing view of the field is that the ubiquitin ligases (and their adapters) that operate on nutrient and ion transporters in the endo-vacuolar system recognize open versus closed conformations; these are well ordered, physiological conformations, not misfolded states. The reviewers suggest that you present your study in terms of "regulated ubiquitination" or a related term. The existence of an "endosomal protein quality control" system that harnesses the DSC complexes could be incorporated appropriately in the Discussion section of the manuscript.

We agree with this comment. It is an overstatement to claim that the Dsc complex is a protein quality control machinery with the evidence we have. Therefore, we toned down our statements in the Abstract and main text. Specifically, 1) we removed “quality control” in the Abstract; 2) instead of stating “An emerging paradigm of the quality control field is that ubiquitin-dependent protein quality control is important for maintaining the integrity of all organelles”, we started our Introduction with “Ubiquitin-dependent protein down-regulation and quality control are important for maintaining the integrity of all organelles”; 3) also in the Introduction, we replaced “protein quality control” with “down-regulation of lysosomal membrane proteins”; 4) in the second paragraph of the Introduction, we changed the sentence to “the Dsc complex might play a role in protein quality control at the downstream organelles of the secretory pathway”; 5) we replaced “Golgi-specific protein quality control complex” with “Golgi-specific E3 ligase complex”; 6) in the last paragraph of the Introduction, we added “probably” in front of “quality control”.

2) The statement that Vld1 and Gld1 "govern the ubiquitination of different organelle membrane proteins independently" should only be a suggestion, as it is not experimentally addressed.

We changed the text to “At these locations, they may govern the ubiquitination of distinct organelle membrane proteins.”

3) The figure legends need to be rewritten to describe strains used and experimental conditions. Currently, it is written more like a Results section.

Thanks for pointing this out. We have re-written the figure legends.